# In Vivo Degradation Behavior of Magnesium Alloy for Bone Implants with Improving Biological Activity, Mechanical Properties, and Corrosion Resistance

**DOI:** 10.3390/ijms24021602

**Published:** 2023-01-13

**Authors:** Shun-Yi Jian, Chiu-Feng Lin, Tung-Lin Tsai, Pei-Hua Wang, Chung-Hwan Chen, Sung-Yen Lin, Chun-Chieh Tseng

**Affiliations:** 1Department of Material Engineering, Ming Chi University of Technology, New Taipei City 24301, Taiwan; 2Center for Plasma and Thin Film Technologies, Ming Chi University of Technology, New Taipei 24301, Taiwan; 3Department of CEO, Metal Industries Research & Development Centre, Kaohsiung 802, Taiwan; 4Combination Medical Device Technology Division, Medical Devices R&D Service Department, Metal Industries Research & Development Centre, Kaohsiung 802, Taiwan; 5Departments of Orthopedics, School of Medicine, College of Medicine, Kaohsiung Medical University, Kaohsiung 80701, Taiwan; 6Orthopaedic Research Center, Kaohsiung Medical University, Kaohsiung 80701, Taiwan; 7Regeneration Medicine and Cell Therapy Research Center, Kaohsiung Medical University, Kaohsiung 80701, Taiwan; 8Musculoskeletal Regeneration Research Center, Kaohsiung Medical University, Kaohsiung 80701, Taiwan; 9Department of Orthopedics, Kaohsiung Municipal Ta-Tung Hospital, Kaohsiung 80145, Taiwan; 10Department of Healthcare Administration and Medical Informatics, Kaohsiung Medical University, Kaohsiung 80701, Taiwan; 11Institute of Medical Science and Technology, National Sun Yat-Sen University, Kaohsiung 80424, Taiwan; 12Department of Orthopedics, Kaohsiung Medical University Hospital, Kaohsiung Medical University, Kaohsiung 80701, Taiwan

**Keywords:** ZK60 magnesium alloy, micro-arc oxidation, EDTA, In-vitro test, biodegradability

## Abstract

This study aimed to establish a surface modification technology for ZK60 magnesium alloy implants that can degrade uniformly over time and promote bone healing. It proposes a special micro-arc oxidation (MAO) treatment on ZK60 alloy that enables the composite electrolytes to create a coating with better corrosion resistance and solve the problems of uneven and excessive degradation. A magnesium alloy bone screw made in this way was able to promote the bone healing reaction after implantation in rabbits. Additionally, it was found that the MAO-treated samples could be sustained in simulated body-fluid solution, exhibiting excellent corrosion resistance and electrochemical stability. The Ca ions deposited in the MAO coating were not cytotoxic and were beneficial in enhancing bone healing after implantation.

## 1. Introduction

Magnesium (Mg) alloys are used to make several artificial human body parts and implants, such as substitutes for hard tissue replacement, fracture healing aids, and fixation devices, because of their light weight, excellent mechanical properties, biocompatibility, and biodegradability [1,2,3,4,5,6,7,8,9,10,11,12,13,14]. Implanted devices often suffer from corrosion because they are exposed to the surrounding body fluids, which typically have high ionic strength. The concentrations of chloride, potassium, and sodium ions are relatively high and may cause a simultaneous electrochemical reaction between the implanted Mg alloys and the surrounding fluids. Mg alloy corrosion may sometimes release ions into body fluids and induce allergies, inflammation, diseases, or cancer [15,16,17]. Moreover, the difference in the electrochemical potential of multiple compositions of Mg alloys may cause more significant electrochemical corrosion [18,19].

It is of great interest for scientists in the fields of medicine and biomaterials to understand the interactions between bone tissues and implant materials. Improvements in the biocompatibility, durability, and longevity of orthopedic implants from the perspectives of patient comfort, mobility, and functionality are of widespread benefit to implant recipients. An important class of biomaterials is bioceramics, such as glass ceramics, Al_2_O_3_, ZrO_2,_ and Ca-P. The advantage of these materials over metals is their corrosion resistance. Calcium phosphate (Ca-P) has been used for over 30 years in clinical applications. The first calcium phosphate material used was a bone substitute. A triple calcium phosphate compound used in a bony defect has been reported to promote osteogenesis [20]. Hydroxyapatite (HA) is the most common Ca-P phase relevant for bio-mineralization owing to its close similarity in composition to the bone mineral as a prospective bone substitute [21,22,23,24]. The application of synthetic calcium phosphate coatings is the most effective method for surface modification, which promotes osseointegration of metallic implants with bone tissues [25]. Ca-P is the principal mineral component of bone tissue, and the presence of several calcium phosphate compounds in the coating provides the possibility of controlling its biochemical activity by changing the phase composition, and the rate of resorption of the coating material must correspond to the rate of tissue restoration [26].

In our previous study [27], we successfully designed a micro-arc oxidation process (MAO) coating with good anti-corrosion performance on the AZ31 alloy. MAO is an effective approach to improve the properties of Mg and its alloys. The process combines electrochemical oxidation with high-voltage spark treatment in an alkaline electrolyte, resulting in the formation of a physically protective oxide film on the metal surface to enhance wear and corrosion resistance as well as prolong the component lifetime [28]. In practice, a thin oxide film on the implant surface, which forms a moist environment, plays an important role in the bioactivity and corrosion behavior of metal implants [29]. In a recent study, thin calcium phosphate layers incorporated into MAO on titanium alloys were studied using the ion-beam deposition method, which enhanced osseointegration [30]. In recent decades, almost all known deposition techniques have been applied to deposit Ca-P coatings. Widely used deposition techniques include plasma spraying, electrostatic spray deposition, sol-gel deposition, and radiofrequency magnetron sputtering [31]. Of course, all of these techniques have their benefits and limitations. Some recent studies have been published on MAO treatment with Ca- and P-containing electrolytes to improve the biocompatibility of Mg alloys [32,33]. Moreover, MAO treatment is an environmentally friendly technology if non-toxic electrolytes are selected [34]. However, no bioactivity investigation has tested whether the application of this MAO treatment will enhance bone regeneration and osseointegration capability in vivo in animal implants.

Previous studies mentioned that some elements used to produce these implants might have toxicity issues. For example, an in vivo test reported by Laing et al. [35] indicated that Ni, Co, Cr, Fe, Mo, V, and Mn are toxic elements because of the unfavorable tissue responses between the metallic implants and rabbit muscle. Elshahawy et al. [36] also evaluated various commercial biomedical alloys using in vitro testing; Cu^2+^, Ni^2+^, and Be^2+^ were identified as toxic ions in fibroblast cell culture. Calin et al. [37] summarized potentially harmful and non-toxic elements for biomedical implants. In our previous study [27], Mg alloy (AZ31) and aluminum (Al) were dissolved. Al is a bone growth inhibitor, a possible cause of Alzheimer’s disease, and is cytotoxic [38]. Cytotoxicity often depends on the ionization tendency of the metals. Highly corrosive materials in the body may release cytotoxic ions and cause cell apoptosis and necrosis after long-term use [39]. Therefore, ZK60 Mg alloys were used as experimental materials in this study. However, few studies have attempted to determine whether ZK60 alloy application enhances rabbit bone regeneration and simultaneously improves mechanical properties, biocompatibility, and corrosion resistance. In this study, we aimed to establish a surface modification technology for ZK60 alloy implants that can uniformly degrade over time and promote bone healing, as well as break through the current technical barriers of uneven degradation and excessive degradation of medical magnesium alloys. The biological activity of the ZK60 alloy was increased by the addition of Ca and P ions in the MAO electrolyte. In addition, as a chelating agent, EDTA was able to improve the calcium content in MAO coatings. An electrolyte containing Ca and P and able to promote osseointegration was deposited in the anodic oxide film during the MAO process. A magnesium alloy bone screw made in this way was able to promote the bone healing reaction after implantation in rabbits. Whether the difference in their bone regenerative potential had an impact on their physical properties and biological activity will be further investigated in our following study.

## 2. Results and Discussion

### 2.1. Characteristics of the MAO-Coated ZK60 Plates

#### 2.1.1. Microstructural Observations

The morphology and cross-sectional microstructure of MAO-coated ZK60 and MAOCa-coated ZK60 are shown in Figure 1. The MAO coating formed without calcium addition had smaller pores and was distributed relatively regularly (Figure 1a). It was found that the thickness of the MAOCa coating was thicker than that of the MAO coating (16.3 ± 1.8 μm vs. 15.7 ± 1.4 μm), as shown in Figure 1b,d. In other words, a thicker coating can be obtained with a calcium compound containing MAO. This is consistent with previous research, where more Ca^2+^ and PO_4_^–3^ ions were added to the MAO treatment solution, which increased the coating thickness [40].

Surface roughness is an important factor affecting the structural integrity of a surface. Figure 2 shows the average roughness (*S_a_*) of the MAO- and MAOCa-coated samples. *S_a_* is a dispersion parameter established as the mean of the absolute values of surface departure above and below the mean plane within the sampling area. The *S_a_* values of the MAO and MAOCa coatings were 0.293 and 0.355 μm, respectively, as shown in Figure 2. This means that the MAO coating was flatter than the MAOCa coating. This result was consistent with the SEM morphology (Figure 1a,c).

#### 2.1.2. Chemical Composition

The chemical composition of both MAO-treated ZK60 samples was analyzed by EDS and SEM. Table 1 shows the EDS analysis results for the different MAO-coated samples on the ZK60 Mg alloys. The EDS results in Table 1 show O, Mg, Na, Si, and P. Among them, the P content in the MAOCa coating was greater than that in the MAO coating, and the Ca content in the MAOCa coating was approximately 4%. In other words, the results clearly show the formation of Ca and P compounds by the MAO process in this study.

#### 2.1.3. Corrosion Resistance

Figure 3 shows the polarization curves of the bare ZK60 and MAO- and MAOCa-coated samples in the SBF solution. The corrosion potential (*E_corr_*) is used to evaluate the driving force for bio-corrosion, and a system with a higher *E_corr_* indicates that more energy is required to initiate the corrosion reaction [41,42,43]. The *E_corr_* values of the bare ZK60, MAO coating, and MAOCa coating were −1.659 V, −1.642 V, and −1.61 V, respectively. The corrosion potential is mainly influenced by the composition. This also shows that the corrosion resistance of both coatings was better than that of bareZK60. Corrosion current density (*i_corr_*) is another important parameter for determining the activity of the corrosion reaction. The *i_corr_* suffers a structural effect, which is utilized to evaluate whether the protective and denser passive layers were formed on the material surface. The *i_corr_* values of bare ZK60, the MAO coating, and the MAOCa coating were 8.73 × 10^–4^ A/cm^2^, 2.03 × 10^–6^ A/cm^2,^ and 1.98 × 10^–6^ A/cm^2^, respectively, indicating that both MAO-coated samples were capable of generating a more protective and compact passive layer on the surface in the SBF solution than bare ZK60. The above results show that both MAO-treated processes degrade the corrosion reaction and retard corrosion activity.

Although the corrosion resistance of both MAO-treated samples could be greatly improved, the corrosion current density was approximately the same. To prove the superior corrosion resistance of the MAOCa coating compared to that of the widely applied MAO coating in biomedical implants, hydrogen evolution tests were conducted. Figure 4 shows the hydrogen evolution of the bare ZK60 and MAO- and MAOCa-coated samples in the SBF solution. It showed lower corrosion rates for the MAOCa-coated samples than for the bare ZK60 and MAO-coated samples in the SBF solution. This is because Ca^2+^ and PO_4_^–3^ ions can reduce the corrosion rate to protect biodegradable medical materials [40,44,45,46]. Figure 5 shows the thickness measurements for both MAO-coated ZK60 samples, recorded every 1 week during long-term immersion in test solutions. Thickness measurement was conducted after immersion in the SBF solution for different periods of up to 13 weeks. It was found that the MAO-coated ZK60 film thickness changed from 15.7 μm to 3.6 μm after immersion for 13 weeks in SBF solution, the film thickness decreased by about 12.1 μm, and the corrosion rate was about 0.93 μm/week. In the same way, the MAOCa-coated ZK60 film thickness changed from 16.3 μm to 6.7 μm after immersion for 13 weeks in SBF solution, the film thickness decreased by about 9.6 μm, and the corrosion rate was about 0.74 μm/week. In short, the MAOCa-coated sample had a great deal of corrosion resistance potential in the SBF solution in the long term. These results show that our synthetic MAOCa-coated sample is suitable for long-term implantation in the corrosive environment of body fluids.

The SST is a rigorous test to demonstrate differences in the corrosion resistance of the specimens. Figure 6 shows the results of the 24 h SST for the bare ZK60 and MAO- and MAOCa-coated samples. When bare ZK60 was subjected to SST for 24 h, the corroded area fraction was >80%. In contrast, there were no rust spots on the MAO-treated samples, which means that they did not affect the corrosion resistance of the additives of Ca^2+^ and PO_4_^–3^ ions in the MAO treatment. This result is consistent with the potentiodynamic polarization and hydrogen evolution measurements. In summary, we successfully created a MAOCa-coated sample with good corrosion resistance by using an electrolytic mixture of Ca_3_PO_4_ and EDTA solutions. MAO coatings were successfully produced with a less porous and denser microstructure and, therefore, exhibited enhanced corrosion resistance.

#### 2.1.4. Biological Activity Enhancement

XPS was performed to further investigate the biological activity and surface chemical composition of the ZK60 plates, as shown in Table 2. Table 2 shows the XPS elemental analysis at 10 nm from the surfaces of the bare ZK60 and MAO-coated ZK60 plates before and after immersion in the SBF solution. Both phosphorus and calcium ions in the MAO oxide film increased after immersion in the SBF solution, which was better than that of the untreated sample. In particular, in the MAOCa treatment (bioactivity) sample after immersion in SBF for 48 h, the elemental analysis of the surface layer down to 10 nm by XPS showed that the calcium and phosphorus ions increased the most. Thus, phosphorus- and calcium-containing components were deposited on the surface of the sample immersed in SBF media after MAOCa treatment of ZK60. To further understand the composition of the oxide film and surface deposits after immersion in the SBF solution, SEM surface morphology and EDS line scan analyses were carried out, as shown in Figure 7. EDS analysis showed that it contained phosphorus and calcium, which confirmed that the MAOCa-coated ZK60 could quickly induce the adhesion and deposition of phosphorus and calcium after immersion in SBF for 48 h. Phosphorus and calcium were only deposited in the corrosion deposits on the surface of the oxide film.

### 2.2. Mechanical Properties of the MAOCa-Coated ZK60 Bone Screws

#### 2.2.1. Results of the Three-Point Flexure Test

Generally, scaffolds with sufficient mechanical strength can maintain a suitable matrix for cell ingrowth, nutrient transport, and physiological loading support. Cells and tissues can infiltrate and grow within the macrostructure and are especially critical for bone reconstruction [47,48]. The MAO process produces porous and uniform oxide coatings with complex geometries on implant surfaces (Figure 1). This porous structure and rough morphology have been proven to increase the mechanical interlocking of tissue and implants, and the Ca-P layer enhances the initial cellular response owing to its high osteoconductivity and bioactivity [49]. Thus, to determine the mechanical properties of the MAOCa-coated ZK60 Mg alloy bone screws, a three-point flexure test was used for mechanical testing. Although this test can provide the results of flexural stress and flexural modulus, this study investigated whether the MAOCa coating of the bone screw cracked after stress. Figure 8 and Table 3 show that the ZK60 Mg alloy bone screws treated with MAOCa were subjected to three-point bending measurements, and cracks in the oxide film under different displacements were observed by SEM. Through SEM observation, it was shown that the oxide film of ZK60 Mg alloy bone screws began to have obvious cracks when they were pressed down by 0.75 mm (Figure 8e). Therefore, the pressing distance was reduced to 0.5 mm, and the results showed that there were no cracks (Figure 8d). These defects were due to the flexure testing results, as it can be assumed that these cracks will extend by enduring a greater force and cause film peeling. It also means that a good implanted material must be able to restrain stress and decrease debris release after long-term use, which stimulates the immune system and inflammatory responses [50].

#### 2.2.2. Adhesion Test Analysis

To understand the adhesion characteristics of the MAOCa-coated ZK60 Mg alloy bone screws, each sample was locked into a pre-drilled D3 synthetic bone, and the adhesion of the oxide layer was observed. In the screw-in experiment, when the hole is 2.2 mm, the screw-in force will be greater than 35 N, and the screw head will be broken or twisted; therefore, the relevant experiment was carried out with a pre-drilled hole of 2.3 mm, as shown in Figure 9a. The surface morphology of the ZK60 Mg alloy bone screw was observed using SEM, and there was no obvious peeling or cracking of the oxide film after screwing, as shown in Figure 9b.

#### 2.2.3. Locking Force Analysis

Although the MAOCa-coated ZK60 showed good electrochemical properties (Figure 3) and biological activity (Figure 7) in SBF solution, studies of its mechanical properties are still rare. Hence, it was necessary to pay more attention to the mechanical properties combined with biological activity, such as immersion in an SBF solution. MAOCa-coated ZK60 Mg alloy bone screws with dimensions of 5 mm × 30 mm were used, and a bone screw locking force test was performed. We screwed the SBF non-immersed and immersed bone screws into pre-drilled and tapped D3 synthetic bone (4.8 mm and a force of 3.5 kg) for 6 and 10 weeks. Then, they were locked into the D3 synthetic bone to 20 mm with the ASTM F543 standard and fixed. Finally, the samples were stretched upward at a speed of 5 mm/min until the ZK60 Mg-alloy bone screws were pulled out of the D3 synthetic bone. The results showed that after 10 weeks of immersion in SBF, the ZK60 Mg alloy bone screw retained 84% (213 N) of its original locking force (251 N), as shown in Table 4. Figure 10 shows that the surface morphology of the ZK60 Mg alloy bone screws and the corrosion products on the surface tended to increase with immersion time in the SBF solution. After the locking force test, no deformation was observed on the surface.

### 2.3. Animal Experiments and In Vitro Cell Test

#### 2.3.1. Biocompatibility Analysis

Biocompatibility is a term that describes the interactions between biomaterials and biological systems that do not have toxic or injurious effects on biological systems [51]. Thus, cytotoxicity testing was used to evaluate the death of living cells in the designed environment. Figure 11 shows the results of the in vitro cytotoxicity test—MTT assays of three cell cultures with the control species, bare ZK60, MAO-coated, and MAOCa-coated specimens after 24 h of immersion in the medium. A blank control was used as the standard for cell viability testing. Ions released into the culture medium could be the major factor causing cell death. The light blue, pink, and yellow bars represent the cell viabilities of the bare ZK60, MAO-coated, and MAOCa-coated specimens, respectively, under the same experimental conditions. The cell viability for each group was higher than 80%, indicating only mild cell death in the in vitro cytotoxicity test. In vitro cell viability and proliferation assays showed that the MAOCa-coated specimens maintained a certain degree of cell viability with minimal cytotoxicity.

To understand the effect of surface modification on the osteoblast-like MG-63 cell viability, the bioactivity of bare ZK60, MAO-coated, and MAOCa-coated specimens was studied using human osteosarcoma (MG-63) cells in MTT cell proliferation tests. Figure 12 shows the cell proliferation rates for MG-63, where the Ca and P ion release numbers for 1, 3, and 7 days are shown. On day 1 of cell proliferation, more Ca and P ions proliferated on the MAOCa-coated specimen compared to the other specimens. The results for day 7 showed that more Ca and P ions proliferated on the MAOCa-coated specimens compared to the untreated and MAO-coated specimens. In particular, the MAOCa-coated specimen exhibited the best cell performance, which was the most remarkable. This indicates that the MAOCa coatings enhanced osteoblastic cell activity and improved bone healing of surrounding tissues after implantation owing to the presence of the Ca-P compound, which was similar to previous studies [30,52,53].

#### 2.3.2. Radiological Examination

To gain further understanding of the relationship between Ca-P coatings and their evident properties as an implantation bed, ZK60 bone screws were implanted into the leg bones of rabbits. Figure 13 presents photo images showing the corresponding X-ray images after surgical operations to implant various ZK60 bone screws (untreated, MAO, and MAOCa-treated) into rabbit tibias. The rabbits were implanted without infection. They recovered well and were fed under intensive care. The physical structures of the three ZK60 bone screws treated with nothing, MAO, or MAOCa were completed after 1 week of implantation, as shown in Figure 13a. However, air cavities were created, representing the phenomenon of existing material degradation (Figure 13b). The X-ray images showed no radiolucent line around any of the three ZK60 bone screws, indicating that there was no local inflammation. Therefore, inflammation-induced osteolysis was prevented and was not observed in the X-ray images. Two weeks after implantation, air cavities were present but not expanded (Figure 13c). After 4–6 weeks of implantation, the formation of air cavities slowed down, but the untreated ZK60 bone screw corroded and gradually disappeared (Figure 13d,e). Finally, the treated ZK60 bone screws were rigidly fixed to the bone after 8 weeks of implantation, which demonstrated that there was no severe degradation (Figure 13f).

#### 2.3.3. Micro-CT Scanning

Figure 14 shows micro-CT images of the untreated, MAO-treated, and MAOCa-treated ZK60 bone screws 12 weeks after implantation, which were used to further evaluate the bone growth and growth properties of the implanted screws. Compared with X-ray detection (Figure 13), the bone screw implantation method in this section involved implanting the bone screws from each process into rabbit bone separately. After 4 weeks of implantation, the lower part of the untreated ZK60 bone screw began to corrode, while the others exhibited no corrosion phenomenon, as shown in Figure 14a,d,g. The untreated ZK60 bone screw was completely corroded after 12 weeks of implantation (Figure 14c). Similarly, the MAO-treated ZK60 bone screw also suffered severe corrosion 12 weeks after implantation (Figure 14f). The most interesting finding in this study was that the *μ*CT images revealed that the bone grew around the MAOCa-treated ZK60 bone screws, and the osteotomy sites healed well (Figure 14g). This indicated that the sample exhibited superior osteoconduction properties. Simultaneously, we identified some newly formed bone tissues that dominated the repaired space. As the implantation time increased, the bony tissues gradually transformed into bone and bone screws without active signs of degradation throughout the experimental period, as shown in Figure 14h. Most of the newly formed bones were also in close contact with the MAOCa-treated ZK60 bone screws and were distributed evenly throughout the repaired space. The addition of bone grafting materials could create more space and mediate osteogenesis [50,54]. In this study, MAOCa treatment induced more bone tissue growth. 

Biocompatibility is the ability of a prosthesis implanted in the body to exist in harmony with the tissue without causing deleterious changes [51]. Hence, biomaterials with good biocompatibility are more suitable for biomedical applications [55]. After materials are implanted in vivo, body fluid contact begins when the pH of plasma and tissue fluid ranges from 7.35 to 7.45. The concentrations of chloride ions in the plasma were 113 and 117 mEq/L, respectively. Such high chloride-ion concentrations can readily corrode implant alloys. In addition, buffer solutions containing amino acids and proteins in the body accelerate the corrosion of implant alloys [56,57]. After patients undergo orthopedic surgery, the pH in the body drops to 5.2 and then rises back to 7.4 within 2 weeks [58]. This large difference in pH causes the denaturation of proteins to further corrode implant alloys. Figure 15 shows the blood tests of the untreated, MAO-treated, and MAOCa-treated ZK60 bone screws 12 weeks after implantation, which were used to further evaluate the physiological and biochemical states. The green lines in Figure 15 represent the standard values for GOT, GPT, serum creatinine, and serum UREA, respectively. Blood analysis of ZK60 bone screw implantation for 12 weeks revealed that GOT and GPT levels were significantly higher in the untreated sample, and MAOCa-treated samples could effectively alleviate physiological changes. This indicated that there was inflammation and a significant corrosion reaction caused by the untreated ZK60 bone screw implanted in the in vivo test. A possible reason may be that blood flow in the bone marrow may bring the released ions away from the implantation site and cannot eliminate the accumulation of released ions from the untreated ZK60 bone screw. In contrast, the low levels of various blood tests in the rabbits after 12 weeks of implantation demonstrated that there was no obvious inflammatory reaction caused by the MAOCa-treated ZK60 bone screw implant. The in vivo test showed that MAOCa-treated ZK60 bone screw implantation had good biocompatibility, and its enhanced osseointegration is probable for biomedical purposes.

## 3. Experimental Section

### 3.1. Preparation of Specimens

A piece of Mg–Zn–Zr alloy (ZK60 Mg alloy) with dimensions of 50 mm × 50 mm × 2.0 mm was used as the metallic substrate. Before MAO treatment, all plates were mechanically ground using SiC papers of up to 2000 grit to ensure the same surface roughness. The plates were ultrasonically cleaned in acetone, rinsed with deionized water, and dried in a stream of hot air at 60 °C. The process of preparing MAO coatings on the ZK60 magnesium surface was carried out on a pulse power generator (MIRDC) with work at a current density of 250 mA/cm^2^, the duty cycle of 60%, and an electrical frequency of 500 Hz for 10 min [27]. The water bath made of stainless steel had a water capacity of 45 L. A glass beaker containing 1 L of electrolyte was placed inside the water bath. A stainless-steel plate was placed inside the glass to serve as the cathode, and a ZK60 alloy substrate was used as the anode in the process. Table 5 lists the composition of the MAO treatment, including with/without calcium compound addition, in which the entire treatment procedure involving the recirculation of cold water was maintained at 25 °C.

### 3.2. Characterization

Microstructure images of the different MAO coating specimens on both the plane and cross-sectional surfaces were obtained using a scanning electron microscope (SEM, JEOL JSM-IT100). ImageJ software was used to process the stored SEM images for the quantification of pore characteristics (number and average size of pores). Digital 3D white-light interferometry (Chroma 7503, Taoyuan, Taiwan) was used to measure the surface roughness of the different MAO coating specimens. In the hydrogen evolution method, the amount of dissolved Mg can be estimated from the volume of hydrogen evolution as a result of the corrosion reaction [59,60]. X-ray photoemission spectroscopy (XPS, PHI 5000 Versa Probe, Kanagawa, Japan) was used to analyze the surface compositions of the MAO films. Owing to the overlapping characterization, the compositions of the as-deposited coatings were checked by XPS instead of EDS; consequently, the detection depth of XPS was more adequate than that of EDS. Additionally, the composition of the passive layer formed on the surface was analyzed after immersion in simulated body fluid (SBF) for 2 days. For the XPS spectra, all of the binding energies were calibrated using the C 1s peak at 284.8 eV. Mechanical properties were measured using a biaxial servo-hydraulic machine (MTS Mini Bionix II 858). The bone screw was manufactured as a tensile specimen, followed by the design [40].

### 3.3. Electrochemical Measurements and Corrosion Test

The electrochemical behavior and bio-corrosion properties of the substrate and MAO-coated samples in SBF (Hank’s solution, pH 6.5) at 37 °C were tested using potentiodynamic polarization tests, which were conducted using an Autolab PGSTAT30 potentiostat-frequency analyzer. A standard three-electrode system was used in this study. A saturated calomel electrode was used as the reference electrode, and all of the potentials were expressed with respect to this electrode. A platinum plate was used as the counter electrode, and the specimens were used as the working electrode, with an immersion area of approximately 1 cm^2^. The state of the electrochemical surroundings with specimens had to remain steady in SBF until the open-circuit potential (OCP) changed by no more than 2 mV/10 min before the potentiodynamic polarization measurements. After stabilization for 30 min at the open circuit potential (OCP), the potential scan rate was controlled at 0.5 mV/sec from −300 mV to 500 mV based on the OCP. The corrosion current density (*i_corr_*), corrosion potential (*E_corr_*), and corrosion rate were determined from the anodic polarization plots using the Tafel extrapolation method. The salt spray test (SST) followed ASTM standard B117 and was performed for each coated ZK60 plate, which was placed at a tilted angle of 30° in a chamber containing 5 wt% NaCl fog. After the salt spray test, the percentage of pitting area was examined using the ASTM D610-08 standard. All tests were repeated three times.

### 3.4. Animal Surgery and Implant Harvest

In vitro cytotoxicity tests were performed to evaluate the biological compatibility of the MAO-coated samples. Extraction of teat samples and treatment of mouse lung fibroblast cells (L929 cells) with teat sample extracts were performed according to ISO10993-12 and ISO10993-5, respectively. Cell viability determined by MTT assay showed that the test sample extract had, on average, <30% inhibitory effects on the viability of cells, as examined by SGS Taiwan Ltd (Taipei, Taiwan).

Animal experiments were conducted per the NIH Guide for Care and Use of Laboratory Animals and were approved by the animal ethics committee of Kaohsiung Medical University (NO: IACUC-103052). New Zealand white rabbits (3.5–4.5 kg, Livestock Research Institute, Taiwan) were used as animal models. Before the MAO coated ZK60 screw samples were implanted into the femoral shaft of a rabbit drill, all animals were kept in a single room, fed a dried diet and water ad libitum, and anesthetized with subcutaneous injection of ketamine 40 mg/kg and xylazine 10 mg/kg. At 4, 8, and 12 weeks post-implantation, the rabbits were euthanized humanely with an intravenous overdose of barbiturate (200 mg/kg).

To visualize the samples and analyze images of new bone formation three-dimensionally, the samples were scanned using a micro-computed tomography scanner (*μ*CT, Skyscan 1272, Bruker, Kontich, Belgium) with a high resolution in vivo *μ*CT scanner for preclinical research. A frame average of 3 was employed along with a filter of 0.11 mm X 2 mm copper. The X-ray tube voltage was 100 kV, the exposure time 2050 ms, and the current 100 A. The compiled CT films were viewed and analyzed using NRecon software where a 3-D model was built to determine the quality of bone regeneration. After the implantation, the rabbits were housed in cages individually and monitored by an experienced veterinarian for signs of infection, inflammation, and any adverse reaction. The skin was dissected, and the implantation site and surrounding bone were harvested by a mini-saw. The specimens were fixed in 10% buffered neutralized formalin for 24 h at room temperature and prepared for *μ*CT and histological analyses.

## 4. Conclusions

In this study, MAOCa-coated ZK60 samples were sustained in SBF, exhibiting the best corrosion resistance and electrochemical stability. Animal experiments and in vitro cell tests confirmed that the MAOCa-coated ZK60 samples have strong potential for biomedical applications because of their superior biocompatibility and very low cytotoxicity. This material was demonstrated to be highly biocompatible and osteoconductive when implanted in rabbit bones. In particular, it exhibited enhanced mechanical properties and in vitro Ca-P compound forming abilities in SBF, and provided mechanical stiffness to overcome some of the drawbacks. Moreover, we expect to develop a biocompatible implant system with adequate mechanical properties and improved corrosion resistance using a simple MAO technique to accelerate post-surgery recovery.

## Figures and Tables

**Figure 1 ijms-24-01602-f001:**
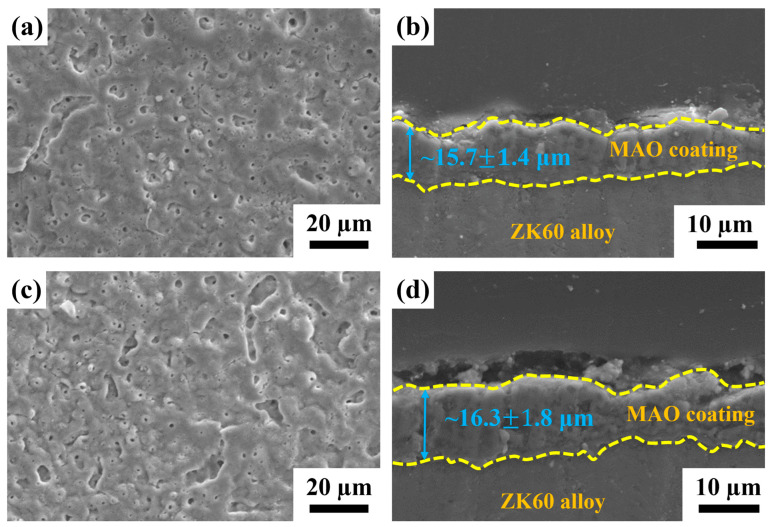
SEM morphology and cross-sectional microstructure of (**a**,**b**) the MAO and (**c**,**d**) the MAOCa coated ZK60 samples.

**Figure 2 ijms-24-01602-f002:**
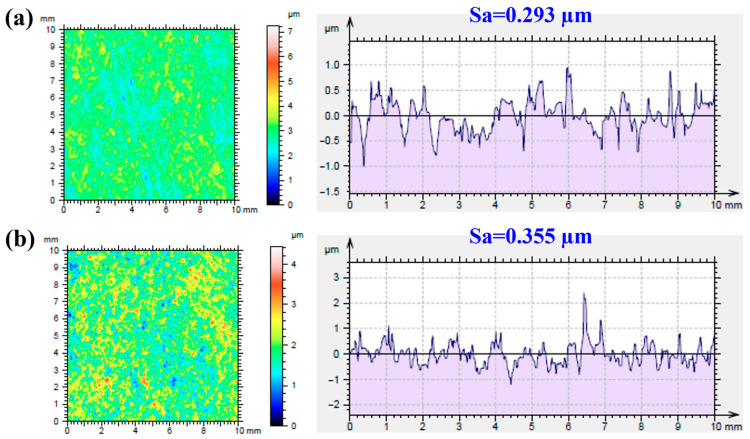
Roughness average of (**a**) the MAO and (**b**) the MAOCa coated ZK60 samples.

**Figure 3 ijms-24-01602-f003:**
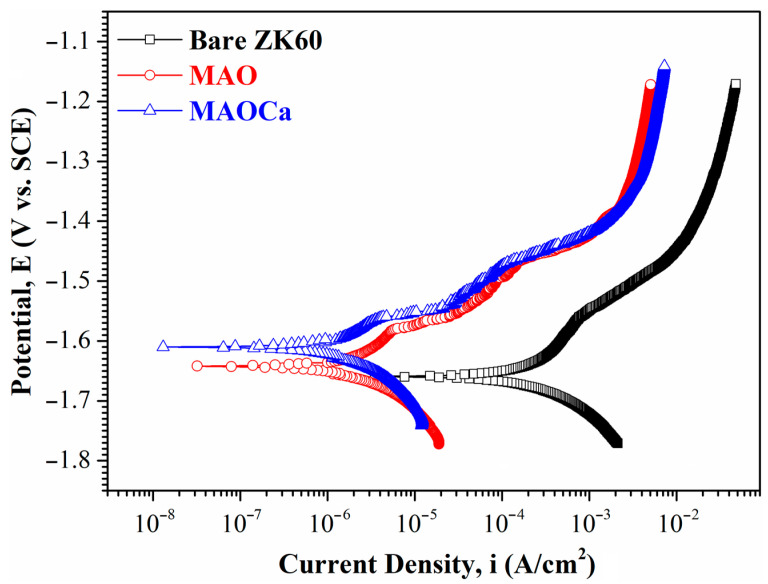
Potentiodynamic polarization curves in SBF solution for the bare ZK60 and samples coated in the MAO and MAOCa treatments.

**Figure 4 ijms-24-01602-f004:**
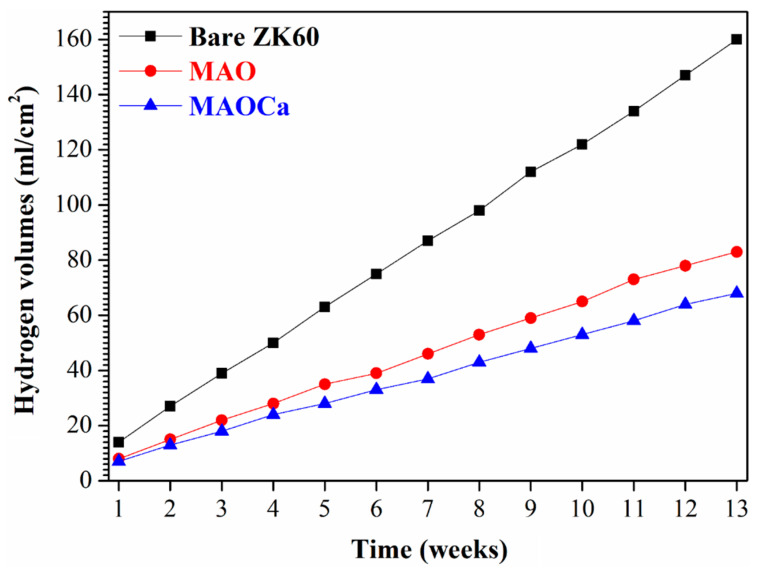
Hydrogen evolution in SBF solution for the bare ZK60 and samples coated in the MAO and MAOCa treatments.

**Figure 5 ijms-24-01602-f005:**
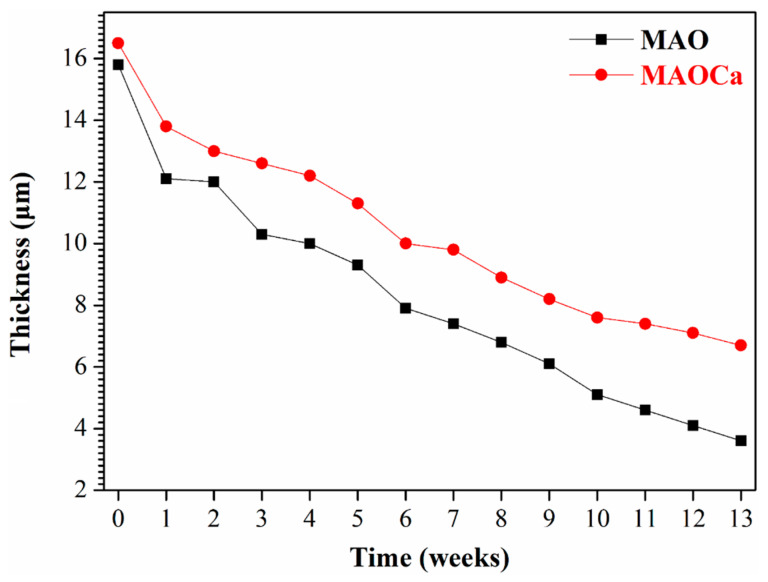
Thickness measurements for the MAO- and MAOCa-coated ZK60, recorded once every week during long-term immersion in SBF solution.

**Figure 6 ijms-24-01602-f006:**
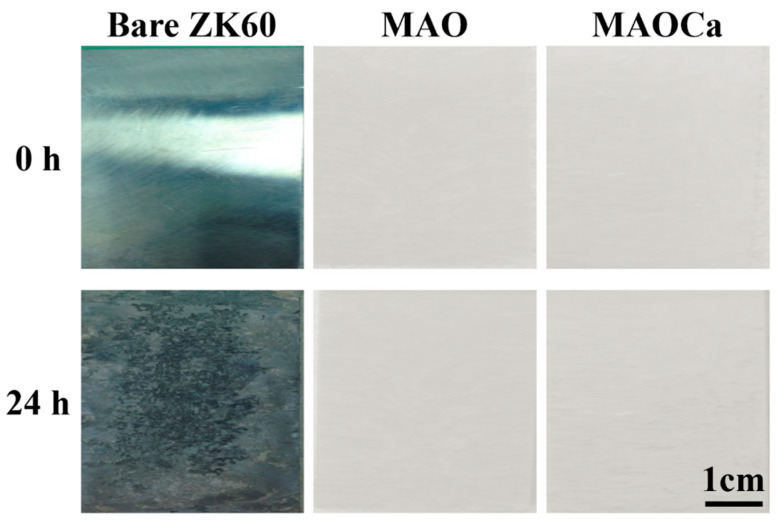
Visual images of the bare ZK60 and samples coated in the MAO and MAOCa treatments after 24 h of the SST.

**Figure 7 ijms-24-01602-f007:**
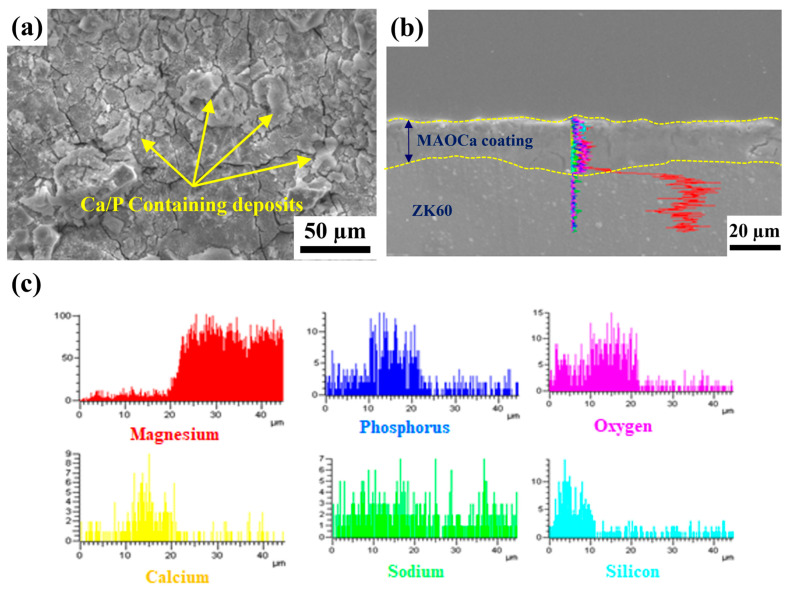
(**a**) SEM morphology and (**b**,**c**) EDS line scan analysis of the MAOCa-coated ZK60 samples.

**Figure 8 ijms-24-01602-f008:**
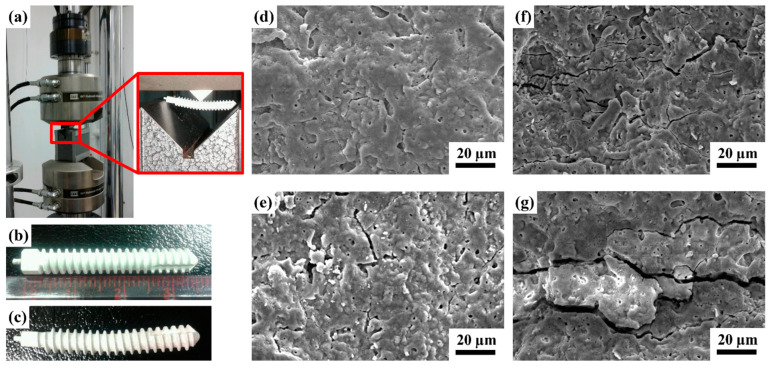
(**a**) Test setup for the three-point bending measurement; visual images of (**b**) before, (**c**) after testing on the ZK60 Mg alloy bone screws treated with MAOCa, and the SEM morphology of the bone screw depression distance (**d**) 0.5; (**e**) 0.75; (**f**) 1.0; (**g**) 2.0 mm.

**Figure 9 ijms-24-01602-f009:**
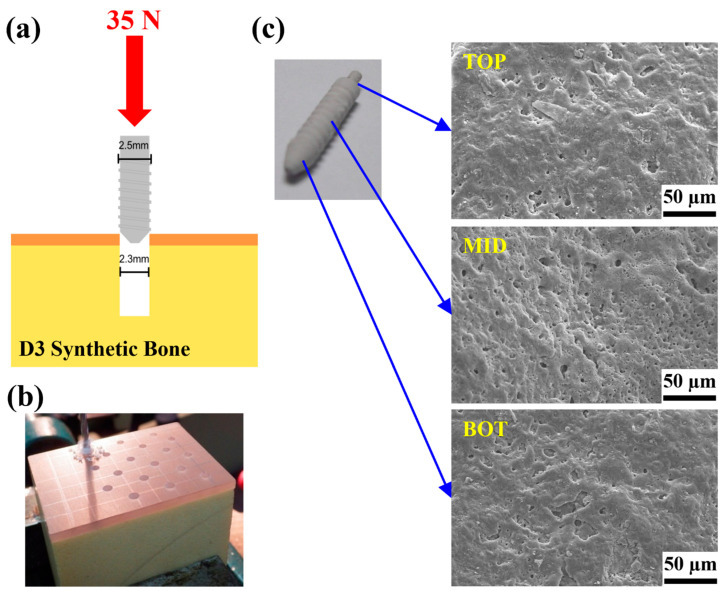
(**a**,**b**) Test setup for the locking force and (**c**) SEM morphology of the ZK60 Mg alloy bone screws treated with MAOCa after the locking force testing.

**Figure 10 ijms-24-01602-f010:**
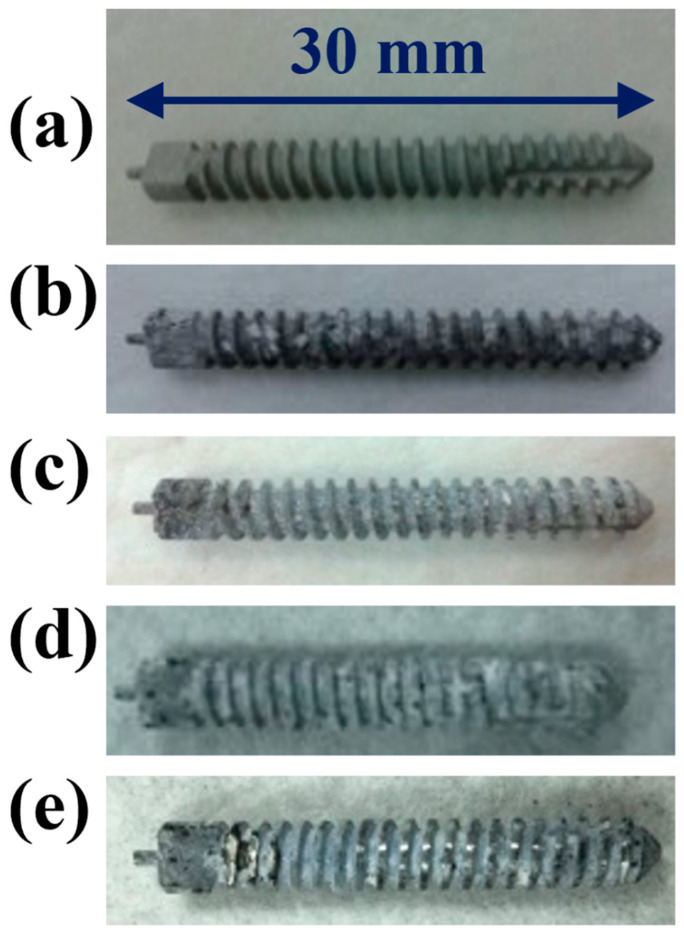
Visual images of the ZK60 Mg alloy bone screw after locking force test: (**a**) SBF non-immersed; (**b**) immersed for 6 weeks; (**c**) immersed for 6 weeks and pulled out of the D3 synthetic bone; (**d**) immersed for 10 weeks; (**e**) immersed for 10 weeks and pulled out of the D3 synthetic bone.

**Figure 11 ijms-24-01602-f011:**
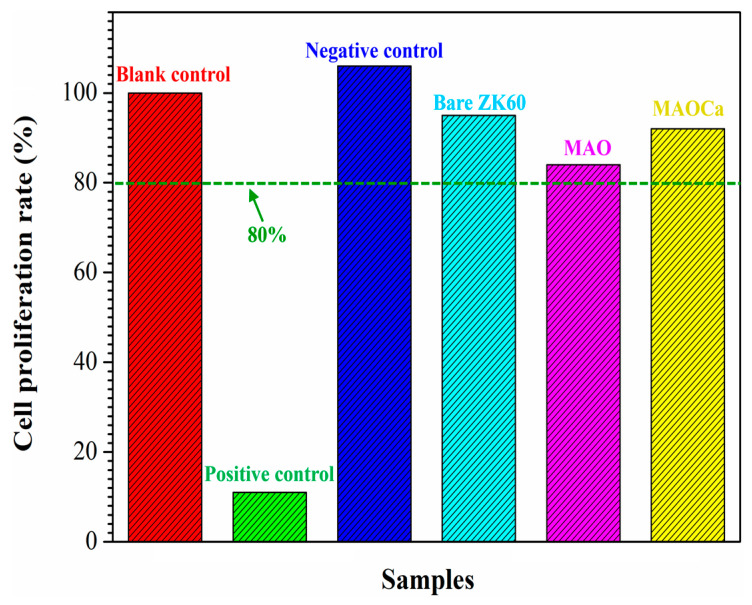
Results of in vitro cytotoxicity test. MTT assays of three cell cultures with the control species, bare ZK60, the MAO-coated and the MAOCa-coated specimens after 24 h immersion in the medium.

**Figure 12 ijms-24-01602-f012:**
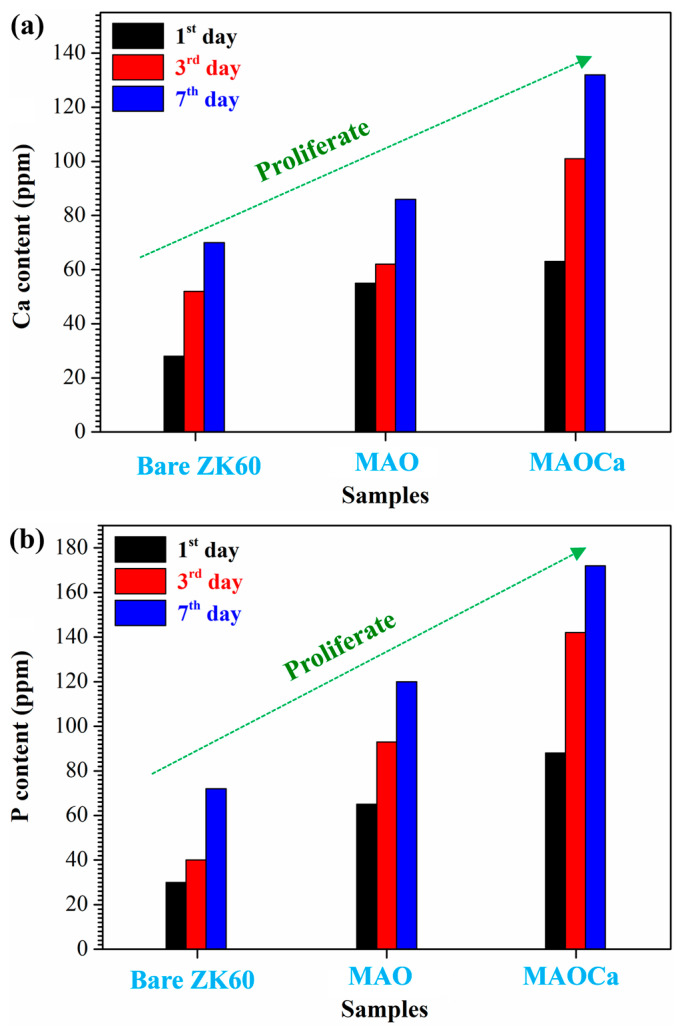
Cell proliferation rates for MG-63: (**a**) Ca and (**b**) P ion release numbers for 1, 3, and 7 days.

**Figure 13 ijms-24-01602-f013:**
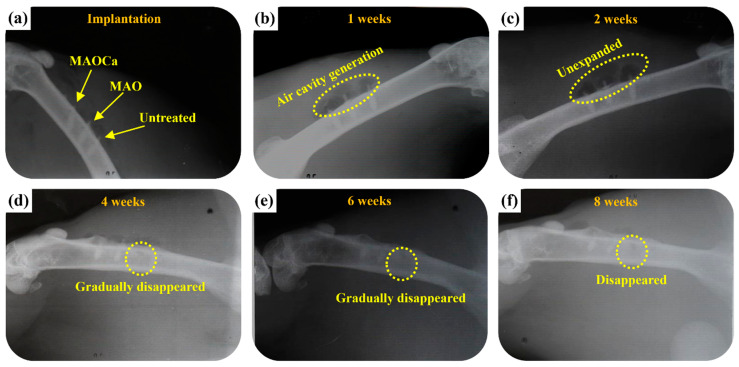
Plain X-rays of the various ZK60 Mg alloy bone screws in the femoral shaft of a rabbit: (**a**) Just implanted, (**b**) 1, (**c**) 2, (**d**) 4, (**e**) 6, and (**f**) 8 weeks.

**Figure 14 ijms-24-01602-f014:**
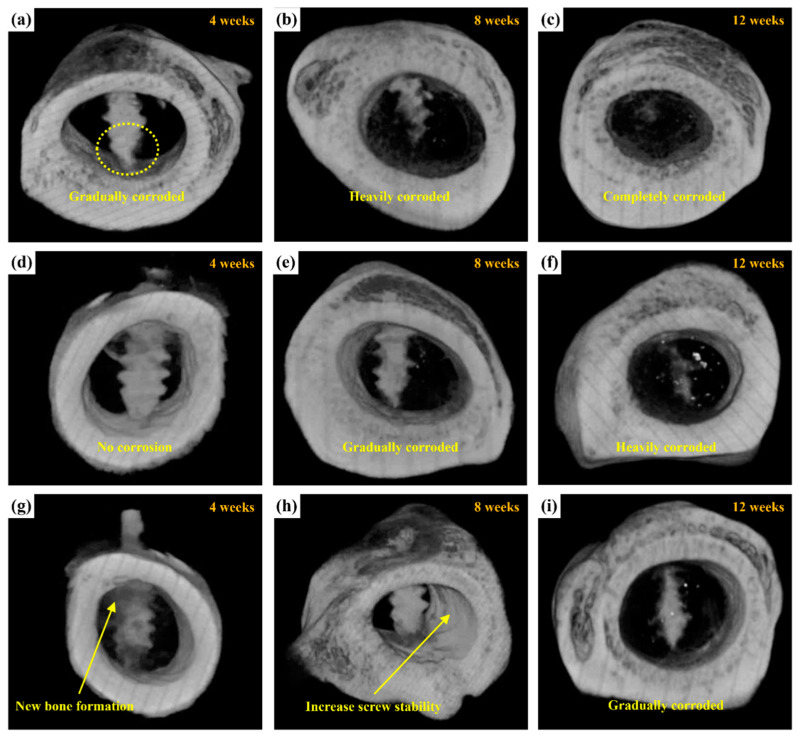
μCT reconstructed images showing the degradation processes of ZK60 bone screw after implantation for 4, 8, and 12 weeks: (**a**–**c**) untreated, (**d**–**f**) MAO-treated, and (**g**–**i**) MAOCa-treated screw.

**Figure 15 ijms-24-01602-f015:**
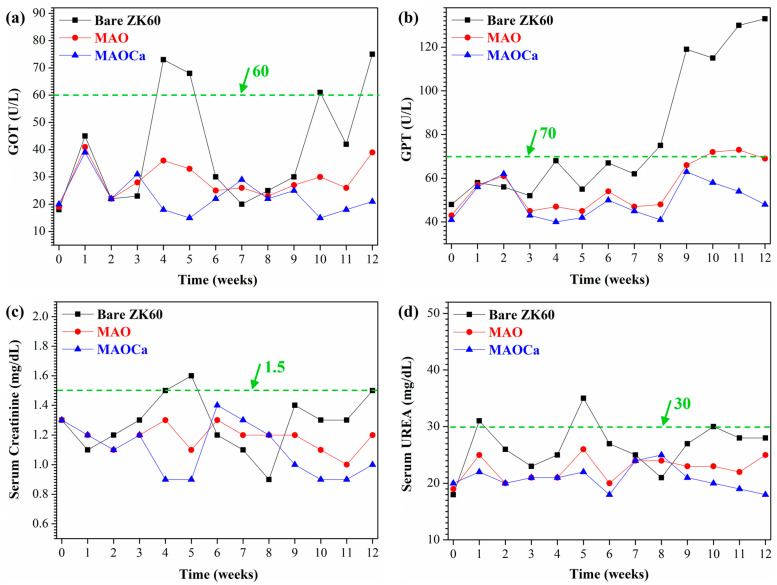
Blood test results for implants of the untreated, MAO-treated and MAOCa-treated ZK60 bone screws 12 weeks after implantation: (**a**) GOT, (**b**) GPT, (**c**) serum creatinine, and (**d**) serum urea.

**Table 1 ijms-24-01602-t001:** EDS analysis of the different MAO-coated samples on ZK60 Mg alloys.

	Element—Atomic%						
	O	Mg	Na	Si	P	Ca	Total
MAO	43.6	42.5	0.6	11.7	1.6	-	100
MAOCa	55.8	28.3	0.2	9.2	2.4	4.1	100

**Table 2 ijms-24-01602-t002:** XPS results for the bare ZK60 and MAO- and MAOCa-coated ZK60 plates before and after immersion in SBF solution.

Elemental Analysis at 10 nm from the Surface	Before Immersion			After 48 h Immersion		
	Bare ZK60	MAO	MAOCa	Bare ZK60	MAO	MAOCa
Ca	0%	0%	4.01%	7.45%	14.6%	18.8%
P	0%	1.82%	2.03%	6.12%	8.92%	11.1%

**Table 3 ijms-24-01602-t003:** Three-point bending measurement of the ZK60 Mg alloy bone screws treated with MAOCa.

	The ZK60 Mg Alloy Bone Screw Depression Distance (mm)			
	0.5	0.75	1.0	2.0
The angle of deformation	3.3°	5°	6.6°	13.1°
Load force	70 N	90 N	201 N	254 N

**Table 4 ijms-24-01602-t004:** Locking force of the ZK60 Mg alloy bone screw.

	ZK60 Mg Alloy Bone Screw
Locking force of SBF non-immersed screw (A)	251 N
Locking force of screw immersed for 6 weeks (B)	233 N
Residual locking force of screw immersed for 6 weeks (B/A)	92%
Locking force of screw immersed for 10 weeks (C)	213 N
Residual locking force of screw immersed for 10 weeks (C/A)	84%

**Table 5 ijms-24-01602-t005:** Electrolyte composition and operating conditions for the MAO treatment.

Name	Na_2_SiO_3_	NaOH	Na_3_PO_4_	Ca_3_PO_4_	EDTA
MAO	60 g/L	70 g/L	20 g/L	-	-
MAOCa	60 g/L	70 g/L	20 g/L	10.5 g/L	7.5 g/L
Electrolyte information	All electrolytes were from ECHO CHEMICAL CO., LTD. (Miaoli County 35145, Taiwan).				

## Data Availability

Not applicable.

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
