# Peer review of "In Vivo Degradation Behavior of Magnesium Alloy for Bone Implants with Improving Biological Activity, Mechanical Properties, and Corrosion Resistance"

_ijms, 2023, doi:10.3390/ijms24021602_

Round 1

Reviewer 1 Report

Although there have been many studies on biodegradable magnesium alloys, this article presents a relatively comprehensive study on the degradation behaviors of MAO-coating with addition of Ca on a ZK60 magnesium alloy, in comparison with the single MAO-coated and un-coated alloy, from in vitro to in vivo, showing beneficial effect on reducing the degradation rate of the alloy. The results were relatively complete, which are valuable for the application. Therefore, the article can be accepted for publication in the present state.

Author Response

Responses to Editor’s and Reviewers’ Comments

Subject:

Ms. Ref. No.: ijms-2093095

Title: In vivo degradation behavior of magnesium alloy for bone implants with improving biological activity, mechanical properties, and corrosion resistance

Dear Editor and Reviewers,

The authors would like to thank you for the possibility given to us to perform a review to our submitted manuscript. The comments are very helpful for us to revise our paper and improve its quality and to make this paper worthy for publication in International Journal of Molecular Sciences.

Additional information and local changes (highlighted in yellow in the manuscript) were done in order to improve the quality of the present paper. The reviewer’s comments were carefully analyzed and used. In the following lines, the performed modifications are presented in the form of an answer to each of the reviewers’ comments.

Reviewer comments: Reviewer #1:

Comment 1:

Although there have been many studies on biodegradable magnesium alloys, this article presents a relatively comprehensive study on the degradation behaviors of MAO-coating with addition of Ca on a ZK60 magnesium alloy, in comparison with the single MAO-coated and un-coated alloy, from in vitro to in vivo, showing beneficial effect on reducing the degradation rate of the alloy. The results were relatively complete, which are valuable for the application. Therefore, the article can be accepted for publication in the present state.

Response:

Thank you for your appreciation. We hope that the revised version will be acceptable for publication in International Journal of Molecular Sciences.

Reviewer 2 Report

In the study, Ca-P coatings were fabricated on magnesium alloys by MAO treatment and in vitro and in vivo properties were systematically evaluated. This study achieves good originality but is insufficient for acceptance in as-received form. I think that this study should be improved in these aspects:

1. The Abstract part should be significantly revised. (1) In my opinion, some important results especially in vivo experimental results should be added, fro example, “A magnesium alloy bone screw made in this way can promote the bone healing reaction after implantation in rabbits”. Meanwhile, some unimportant sentences in present Abstract part should be deleted, for example, “The results demonstrated that a current density of 250 mA/cm2, time of 10 min, frequency of 500 Hz, and a duty cycle of 60% during the MAO process, results in the corrosion resistance significantly increasing after adding bioactive drugs such as EDTA and calcium/phosphorus ions”. (2) The authors emphasized that the used MAO electrolytes were composed of non-toxic chelating agents (EDTA) and bioactive calcium/phosphorus ions, which should be placed in Introduction part. In addition, in Introduction part, the authors should add the characteristics of MAO treatment, for example, MAO treatment is an environmentally friendly technology if nontoxic electrolytes are selected. Some recent works about environmental friendly methods or technologies on MAO treatment of magnesium alloys should be added, for example, Jin Qin, Xiaoting Shi, Hongyu Li, et al., Performance and failure process of green recycling solutions for preparing high degradation resistance coating on biomedical magnesium alloys, Green Chemistry, 24 (2022) 8113-8130. (3) The authors think that EDTA belongs to a bioactive substance. In my opinion, as a chelating agent, EDTA can improve the calcium content in MAO coatings but is not a bioactive substance.

2. In Introduction part, the authors wrote “However, little information has been documented on MAO treatment with Ca- and P-containing electrolytes to improve the biocompatibility of Mg alloys”. In fact, Ca-P coatings have become a hot issue within the field of metallic implants. Some recent works on the aspect should be cited, for example, (1) Xiaoting Shi, Yu Wang, Hongyu Li, et al., Corrosion resistance and biocompatibility of calcium-containing coatings developed in near-neutral solutions containing phytic acid and phosphoric acid on AZ31B alloy, Journal of Alloys and Compounds, 823 (2020) 153721. (2) Z.P. Yao, L.L. Li, Z.H. Jiang, Adjustment of the ratio of Ca/P in the ceramic coating on Mg alloy by plasma electrolytic oxidation, Appl. Surf. Sci. 255 (2009) 6724-6728.

3. In “2.1. Preparation of specimens” part, more detail in electrolytes information such as manufacturer, country, and specification needs to be stated. Especially, according to Table 1, 10.5 g/L Ca3PO4 was used in this study. In fact, inorganic calcium salts including Ca3PO4, CaF2, CaCO3 exhibit poor solubility in aqueous solution, which is difficult to reproducibly fabricate high calcium content MAO coatings in alkaline solutions. I wonder what methods in this study were used to improve Ca3PO4 solubility.

4. Except the full name of salt spray test (SST) being written for the first time, its abbreviated name “SST” should be used in the caption of Figure 6.

5. The existing phases of Ca-P compounds are closely related with their bioactivity. For example, hydroxyapatite (HA) is the most common Ca-P phase relevant for bio-mineralization. Therefore, the phase structure of the substrate and MAO-treated samples before and after immersion in SBF solution should be measured by XRD analysis.

6. The sequence in some paragraphs should be changed. In general, in vitro biocompatibility tests should be completed before in vivo experiments. Therefore, “3.3.3. Biocompatibility analysis” should be adjusted into “3.3.1. Biocompatibility analysis”.

7. For magnesium alloys, the corrosion resistance of MAO treated samples is very important and therefore in some papers the influencing factors including chemical compositions, pore size, porosity, coating thickness, bonding strength, and wettability, et al have been widely discussed. Therefore, the main influencing factors on the corrosion resistance in this study should be further discussed in the revised manuscript.

Author Response

Responses to Editor’s and Reviewers’ Comments

Subject:

Ms. Ref. No.: ijms-2093095

Title: In vivo degradation behavior of magnesium alloy for bone implants with improving biological activity, mechanical properties, and corrosion resistance

Dear Editor and Reviewers,

The authors would like to thank you for the possibility given to us to perform a review to our submitted manuscript. The comments are very helpful for us to revise our paper and improve its quality and to make this paper worthy for publication in International Journal of Molecular Sciences.

Additional information and local changes (highlighted in yellow in the manuscript) were done in order to improve the quality of the present paper. The reviewer’s comments were carefully analyzed and used. In the following lines, the performed modifications are presented in the form of an answer to each of the reviewers’ comments.

Reviewer comments: Reviewer #2:

Comment 1:

The Abstract part should be significantly revised. (1) In my opinion, some important results especially in vivo experimental results should be added, fro example, “A magnesium alloy bone screw made in this way can promote the bone healing reaction after implantation in rabbits”. Meanwhile, some unimportant sentences in present Abstract part should be deleted, for example, “The results demonstrated that a current density of 250 mA/cm2, time of 10 min, frequency of 500 Hz, and a duty cycle of 60% during the MAO process, results in the corrosion resistance significantly increasing after adding bioactive drugs such as EDTA and calcium/phosphorus ions”. (2) The authors emphasized that the used MAO electrolytes were composed of non-toxic chelating agents (EDTA) and bioactive calcium/phosphorus ions, which should be placed in Introduction part. In addition, in Introduction part, the authors should add the characteristics of MAO treatment, for example, MAO treatment is an environmentally friendly technology if nontoxic electrolytes are selected. Some recent works about environmental friendly methods or technologies on MAO treatment of magnesium alloys should be added, for example, Jin Qin, Xiaoting Shi, Hongyu Li, et al., Performance and failure process of green recycling solutions for preparing high degradation resistance coating on biomedical magnesium alloys, Green Chemistry, 24 (2022) 8113-8130. (3) The authors think that EDTA belongs to a bioactive substance. In my opinion, as a chelating agent, EDTA can improve the calcium content in MAO coatings but is not a bioactive substance.

Response:

Thank you for your instruction. The correction has been made accordingly. Please see the highlights in ABSTRACT (Line 34-36), INTRODUCTION (Line 84-87, and 107-109), and REFERENCES (34) of the revised manuscript.

Comment 2:

In Introduction part, the authors wrote “However, little information has been documented on MAO treatment with Ca- and P-containing electrolytes to improve the biocompatibility of Mg alloys”. In fact, Ca-P coatings have become a hot issue within the field of metallic implants. Some recent works on the aspect should be cited, for example, (1) Xiaoting Shi, Yu Wang, Hongyu Li, et al., Corrosion resistance and biocompatibility of calcium-containing coatings developed in near-neutral solutions containing phytic acid and phosphoric acid on AZ31B alloy, Journal of Alloys and Compounds, 823 (2020) 153721. (2) Z.P. Yao, L.L. Li, Z.H. Jiang, Adjustment of the ratio of Ca/P in the ceramic coating on Mg alloy by plasma electrolytic oxidation, Appl. Surf. Sci. 255 (2009) 6724-6728.

Response:

Thank you for your instruction. We have been carefully added these important references in the revised manuscript. Please see the highlights in REFERENCES (32, 33) of the revise manuscript.

Comment 3:

In “2.1. Preparation of specimens” part, more detail in electrolytes information such as manufacturer, country, and specification needs to be stated. Especially, according to Table 1, 10.5 g/L Ca3PO4 was used in this study. In fact, inorganic calcium salts including Ca3PO4, CaF2, CaCO3 exhibit poor solubility in aqueous solution, which is difficult to reproducibly fabricate high calcium content MAO coatings in alkaline solutions. I wonder what methods in this study were used to improve Ca3PO4 solubility.

Response:

Thank you for your instruction. We have been carefully added these important references in the revised manuscript. Please see the highlights in Table 1 of the revise manuscript. In addition, Ca3PO4 really cannot be completely dissolved in the treatment solution, but is suspended in the solution. While in the MAO process, Ca3PO4 is like that the composite plating with particles embedded into the metal matrix can enhance the properties of the coating. And, EDTA as a chelating agent, can improve the calcium content in MAO coatings. The mechanism is still under discussion and will not be described in this study. We sincerely hope for your understanding. About the methods of Ca3PO4 solubility, it will be considered in our further study.

Comment 4:

Except the full name of salt spray test (SST) being written for the first time, its abbreviated name “SST” should be used in the caption of Figure 6.

Response:

Thank you for your instruction. The correction has been made accordingly. Please see the highlights in Line 283 of the revise manuscript.

Comment 5:

The existing phases of Ca-P compounds are closely related with their bioactivity. For example, hydroxyapatite (HA) is the most common Ca-P phase relevant for bio-mineralization. Therefore, the phase structure of the substrate and MAO-treated samples before and after immersion in SBF solution should be measured by XRD analysis.

Response:

Thank you for your instruction. We have been carefully re-checked the XRD spectra. In our experiment, Ca-P-related components are amorphous, they are not in crystalline form. So, it’s normal and natural not to be seen by XRD analysis due to its amorphous structure. However, it was needed to have additional analysis such as EDS and XPS to prove the presence of Ca and P compounds at the surface layer. From EDS and XPS results, Ca-P components were confirmed. Please see the Figure 7, Table 2, and Table 3 of the revise manuscript.

Comment 6:

The sequence in some paragraphs should be changed. In general, in vitro biocompatibility tests should be completed before in vivo experiments. Therefore, “3.3.3. Biocompatibility analysis” should be adjusted into “3.3.1. Biocompatibility analysis”.

Response:

Thank you for your instruction. The correction has been made accordingly. Please see the highlights in Page 13-18 of the revise manuscript.

Comment 7:

For magnesium alloys, the corrosion resistance of MAO treated samples is very important and therefore in some papers the influencing factors including chemical compositions, pore size, porosity, coating thickness, bonding strength, and wettability, et al have been widely discussed. Therefore, the main influencing factors on the corrosion resistance in this study should be further discussed in the revised manuscript.

Response:

Thank you for your instruction. The correction has been made accordingly. Please see the highlights in Line 268-271 of the revise manuscript.

Round 2

Reviewer 2 Report

After revision, the manuscript has been improved significantly. I recommend its acceptance for publication after the following respects have been revised.

(1) In Abstract part, “Besides, it is found that the MAO treated samples can sustain in the simulation body fluid solution, exhibiting the best (“the best” should be changed into “excellent”) corrosion resistance and electrochemical stability”.

(2) In Introduction part, the sentence of “Understanding the interactions between bone tissues and implant materials is of great interest to scientists in the fields of medicine and biomaterials” should be changed into “It is of great interest for scientists in the fields of medicine and biomaterials to understand the interactions between bone tissues and implant materials”.

(3) In Introduction part, “MAO is an effective approach to improving (“improving” should be changed into “improve”) the properties of Mg and its alloys”.

(4) “The salt spray test (SST) followed ASTM standard B117 and (“and” should be deleted) was performed for each coated ZK60 plate,…”.

(5) The sentence of “All animals were kept in a single room and fed a dried diet and water ad libitum, anesthetized with subcutaneous injection of ketamine 40 mg/kg and xylazine 10 mg/kg and then the MAO coated ZK60 screw sample were implanted into the femoral shaft of a rabbit drill” should be changed into “Before the MAO coated ZK60 screw samples were implanted into the femoral shaft of a rabbit drill, all animals were kept in a single room, fed a dried diet and water ad libitum, and anesthetized with subcutaneous injection of ketamine 40 mg/kg and xylazine 10 mg/kg”.

(6) The sentence of “That changes the microstructure of the coatings and produced a less porous and denser structure which enhances the corrosion resistance” should be changed into “MAO coatings were successfully produced with a less porous and denser microstructure and therefore exhibited enhanced corrosion resistance”.

(7) “The physical structures of the three ZK60 bone screws treated by none, Mao (“Mao” should be changed into “MAO”), and MAOCa were completed after 1 week of implantation, as shown in Fig. 13(a)”.

Author Response

Dear Editor and Reviewers,

The authors would like to thank you for the possibility given to us to perform a review to our submitted manuscript. The comments are very helpful for us to revise our paper and improve its quality and to make this paper worthy for publication in International Journal of Molecular Sciences.

Additional information and local changes (highlighted in yellow in the manuscript) were done in order to improve the quality of the present paper. The reviewer’s comments were carefully analyzed and used. In the following lines, the performed modifications are presented in the form of an answer to each of the reviewers’ comments.

Reviewer comments: Reviewer #2:

Comment 1:

In Abstract part, “Besides, it is found that the MAO treated samples can sustain in the simulation body fluid solution, exhibiting the best (“the best” should be changed into “excellent”) corrosion resistance and electrochemical stability”.

Response:

Thank you for your instruction. The correction has been made accordingly. Please see the highlights in ABSTRACT (Line 36) of the revised manuscript.

Comment 2:

In Introduction part, the sentence of “Understanding the interactions between bone tissues and implant materials is of great interest to scientists in the fields of medicine and biomaterials” should be changed into “It is of great interest for scientists in the fields of medicine and biomaterials to understand the interactions between bone tissues and implant materials”.

Response:

Thank you for your instruction. The correction has been made accordingly. Please see the highlights in INTRODUCTION (Line 54-55) of the revised manuscript.

Comment 3:

In Introduction part, “MAO is an effective approach to improving (“improving” should be changed into “improve”) the properties of Mg and its alloys”.

Response:

Thank you for your instruction. The correction has been made accordingly. Please see the highlights in INTRODUCTION (Line 73) of the revised manuscript.

Comment 4:

“The salt spray test (SST) followed ASTM standard B117 and (“and” should be deleted) was performed for each coated ZK60 plate,…”.

Response:

Thank you for your instruction. The correction has been made accordingly. Please see the highlights in Line 164 of the revise manuscript.

Comment 5:

The sentence of “All animals were kept in a single room and fed a dried diet and water ad libitum, anesthetized with subcutaneous injection of ketamine 40 mg/kg and xylazine 10 mg/kg and then the MAO coated ZK60 screw sample were implanted into the femoral shaft of a rabbit drill” should be changed into “Before the MAO coated ZK60 screw samples were implanted into the femoral shaft of a rabbit drill, all animals were kept in a single room, fed a dried diet and water ad libitum, and anesthetized with subcutaneous injection of ketamine 40 mg/kg and xylazine 10 mg/kg”.

Response:

Thank you for your instruction. The correction has been made accordingly. Please see the highlights in Line 178-181 of the revise manuscript.

Comment 6:

The sentence of “That changes the microstructure of the coatings and produced a less porous and denser structure which enhances the corrosion resistance” should be changed into “MAO coatings were successfully produced with a less porous and denser microstructure and therefore exhibited enhanced corrosion resistance”.

Response:

Thank you for your instruction. The correction has been made accordingly. Please see the highlights in Line 270-272 of the revise manuscript.

Comment 7:

“The physical structures of the three ZK60 bone screws treated by none, Mao (“Mao” should be changed into “MAO”), and MAOCa were completed after 1 week of implantation, as shown in Fig. 13(a)”.

Response:

Thank you for your instruction. The correction has been made accordingly. Please see the highlights in Line 411 of the revise manuscript.